# Spherulites: How Do They Emerge at an Onset of Nonequilibrium Kinetic-Thermodynamic and Structural Singularity Addressing Conditions?

**DOI:** 10.3390/e24050663

**Published:** 2022-05-09

**Authors:** Jacek Siódmiak, Adam Gadomski

**Affiliations:** Institute of Mathematics and Physics (Group of Modeling of Physicochemical Processes), Faculty of Chemical Technology and Engineering, Bydgoszcz University of Science and Technology, 85-796 Bydgoszcz, Poland; agad@pbs.edu.pl

**Keywords:** spherulites, (poly)crystal formation, complex growing phenomenon, soft condensed matter, nonequilibrium thermodynamics, physical kinetics, entropy production

## Abstract

This communication addresses the question of the far-from-equilibrium growth of spherulites with different growing modes. The growth occurs in defects containing and condensed matter addressing environments of (bio)polymeric and biominerals involving outcomes. It turns out that it is possible to anticipate that, according to our considerations, there is a chance of spherulites’ emergence prior to a pure diffusion-controlled (poly)crystal growth. Specifically, we have shown that the emergence factors of the two different evolution types of spherulitic growth modes, namely, diffusion-controlled growth and mass convection-controlled growth, appear. As named by us, the unimodal crystalline Mullins–Sekerka type mode of growth, characteristic of local curvatures’ presence, seems to be more entropy-productive in its emerging (structural) nature than the so-named bimodal or Goldenfeld type mode of growth. In the latter, the local curvatures do not play any crucial roles. In turn, a liaison of amorphous and crystalline phases makes the system far better compromised to the thermodynamic-kinetic conditions it actually, and concurrently, follows. The dimensionless character of the modeling suggests that the system does not directly depend upon experimental details, manifesting somehow its quasi-universal, i.e., scaling addressing character.

## 1. Introduction

Defects in condensed matter phase are ubiquitous in their appearances and types. As for the point defects, one can mention atomic vacancies and interstitial positions for the abundant atoms covering the corresponding material’s locations. 

As for extended defects, in turn, it is appropriate to list grain boundaries, two-dimensional defects, very characteristic of polycrystals, whether composed of metallic or of (bio)polymeric material. Stepping up one dimension higher, one can address (screw) dislocations and disclinations, to mention but two [1]. 

Spherulites are complex defects emerging in condensed phase. They bear something from the notion of extended defects, but they mean something distinctly more than this. They are typically recognized as an imperfect crystalline phase comprised from radially distributed polycrystal’s needles, crossed at certain non-crystallographic angles with each other, but intermingled with a not-yet-crystallized amorphous phase in between, cf. Figure 1. The most known technique to visualize the spherulites as defects is polarized light microscopy, which yields the famous Maltese cross. Especially, the amorphous, non-radial phase of the object is named a “band defect” the expression of which depends on peculiarities of the system studied [2]. It seems as if it was a process of competition between crystal and amorphous emerging phases that tend to evolve in a diffusional manner, $r(t)\sim t^{1/2}$ (r—arbitrary spherulite’s radius in diffusion-controlled mode; t—time), and then, they change their growing mode toward its mass-convection-like (long times) counterpart, r(t)~t. This implies that the evolution goes with a constant speed [3,4,5] and proceeds with certain massive steps in absorbing the amorphous matter of the system. It looks as if it was that the emerging (poly)crystalline phase imposes a confinement on the diffusion space, yielding an accelerated absorption of the diffusing material onto the crystal phase, with a remnant non-absorbed amorphous phase still available for diffusion to occur. The method of revealing this phenomenon turns out to be the DSC (Differential Scanning Calorimetry), and the underlying process is coalescence of the spherulitic material, also resulting in a structural impingement of the spherulites [6].

An intriguing question that appears sounds: Why the spherulitic evolution changes its mode from the diffusion-like to that mass-convection-like? Amongst many answers to this question, there is at least a pronounced streamline of arguments proclaiming that the growing system of interest is evolving in nonequilibrium thermodynamic boundary conditions [3,7]. Being motivated by the aforepresented and not-answered-in-full experimental observations, in what follows, we are attempting to provide a simple theoretical rationalization that it is convincingly seen in terms of our type of modeling.

To achieve our goal, we shall employ in a natural way a spherical approximation to a conserved-mass deposition, mimicking the spherulitic growth. It seems to be really natural here, because the complex defects called spherulites assume ultimately a sphere-like form. The spherical approximation to the mass conservation law has been presented elsewhere [8,9,10]. However, the essential novelty applied to it is going to rely on a special boundary condition of nonequilibrium character [11]. Another relevant precondition applied to the spherulite’s modeling, especially important for (bio)macromolecular realizations, is that the mass-convective instead of purely diffusive mass transportation conditions are employed decisively to create the spherulite’s evolution. However, the diffusion limit is not completely ruled out, but it is also discussed when considering the onset of spherulitic formations [8,9,10,11]. 

In the subsequent sections, we shall present the model of spherulitic formation, capturing both diffusional and non-diffusional/mass-convection-like competitive modes, and bearing a signature of nonlinear ordinary differential equation (ode), solvable when resorting to its numerical solutions [12]. The presentation of the spherulite-formation model in terms of nonlinear ode with the corresponding initial conditions gives also a chance of its qualitative analysis. It leads to determining spherulite’s characteristic linear dimension valid for both regimes recalled, albeit the radius’ value for the immature spherulite (rather, its prerequisite) at the onset of the spherulitic growth is half the size as it would be for its purely diffusive counterpart. The proposed semi-quantitative model looks fairly manageable to solicit firm-basis addressing conclusions toward spherulites’ formation. It is because this is to a major extent presented in terms of rescaled variables [12], both independent and dependent (the key parameters as well). Thus, all quantities of interest are nondimensional, and the number of the key governing parameters is reduced from five to two, see Section 2. It univocally allows to conclude on the principal features of the proposed modeling, showing up basic signatures of the spherulitic growth [3,4,7], and its, most importantly, inherent passage from diffusional to mass-convection-like limit, a type of structural, nonequilibrium phase transition [6]. 

The article is organized as follows. In Section 2, a kinetic-thermodynamic nonequilibrium model of spherulitic growth is presented, whereas in Section 3 its main results toward the onset of the spherulitic growth are disclosed, and its properties, also certain proposals for legitimate and/or useful extensions, are discussed. Section 4 serves for the main conclusions.

## 2. Spherulites’ Formation in Terms of a Kinetic-Thermodynamic Model

Herein below, let us consider in brief a model of the spherulitic growth that is based on a mass-convection conserved field instead of a diffusion field. We would like to state clearly that, considering the growth of spherulites, here from solution of a certain concentration, we propose our simple approach in which the spherulites are represent by spherical objects. Note that a spherulite is a 3D-system (there also exist some 2D objects commonly known as cylindrulites). The non-equilibrium character of the process can quantitatively be manifested by at least: (i) external concentration field feeding the growing object, (ii) internal boundary condition prescribed at the interface: spherulite-surroundings.

It is worth recalling the following experimental observation: the growth rate v≡dR/dt (R is the spherulite’s radius; R≡R(t), where t is time) is mainly a parametric function of temperature T (the process under study is isothermal) and slightly depends on a particular system of interest. Thus, we may solely expect that asymptotically R∝t. Note that, especially in the long times’ domain, it substantially differs from the well-known relationship $R\propto t^{1/2}$ characteristic for purely diffusion-controlled crystal growth processes, as first uncovered by the perennially alive Mullins–Sekerka approach [13]. This approach assumes that the growth rate depends on local curvatures of the interface, the growing object vs. surroundings. From this, the square root radius vs. time relationship emerges. 

An evolution equation for spherulites can be formulated as follows:(i)the mass conservation law as a fundamental evolution equation for growing spherulite, with an initial condition which is an initial shape (a surface) of the growing spherulite;(ii)specification of the concentration field of the mass-feeding medium and of the spherulite at the interface; it, in general, should be associated with non-equilibrium boundary conditions;(iii)specification of fluxes through the interface and its connection with the concentration field of the corresponding surroundings; in general, it allows to introduce not exclusively diffusive but also others, such as mass-convection fluxes of atoms, molecules, oligomers and aggregates, etc.

It has been shown that an evolution equation for growing objects (like polycrystals) with an ideal or perturbed spherical symmetry has the form:(1)$$\left[C-c\left(R\right)\right]\dot{R}=-\vec{𝚥}\left[c\left(R\right)\right]\circ {\vec{n}}_{0},$$
where $\dot{R}=dR/dt$, C is the object’s density, which may depend on space variables and can generally be of stochastic nature, c(R) stands for concentration of external particles at the surface, $\vec{𝚥}\left[c\left(R\right)\right]$ is the flux of particles outside the object which depends functionally on concentration and ${\vec{n}}_{0}$ is the outer normal to the surface of the object. Both sides of Equation (1) are given in SI units of $\mathrm{kg}/m^{2}s$.

As regarding point (ii), the concentration of the particles at the surface of the growing object is determined by thermodynamic conditions and geometry of the surface. Under assumption of local thermodynamical equilibrium near the interface, it has the form of the Gibbs–Thompson relation. In a more realistic model, the surface is far from equilibrium and its deviation from equilibrium is proportional to the growth velocity of the interface:(2)$$c\left(R\right)=c_{0}\left(1+\frac{2\Gamma }{R}-\beta \dot{R}\right),$$
where $c_{0}$ is the concentration field at a flat interface, Γ is the capillary constant which aims at smoothening out the surface of the growing object and is proportional to the surface tension [14], β is a positive kinetic coefficient, 2/R is twice the mean curvature of the spherical object, and the last term describes a deviation from the thermodynamic equilibrium. When β=0, one gets the well known Gibbs–Thomson condition.

Let us consider point (iii). If the feed of the growing object is purely mass-convective, then:(3)$$\vec{𝚥}\left[c\left(R\right)\right]=c\left(R\right)\vec{v}\left(R\right),$$
where $\vec{v}\left(R\right)$ is a mass-convection velocity. (For the diffusion-limited, Mullins–Sekerka type growth, r.h.s. of Equation (3) assumes a concentration-gradient form [13]).

From the above, a basic evolution equation can be derived:(4)$$\left[C-c_{0}\left(1+\frac{2\Gamma }{R}-\beta \dot{R}\right)\right]\dot{R}=c_{0}v_{0}\left[1+\frac{2\Gamma }{R}-\beta \dot{R}\right],$$
where $R\left(t=t_{0}\right)>0$.

The growth process described by Equation (4) is influenced by five parameters: C, $c_{0}$, Γ, β and $v_{0}$. However, in fact, only two parameters are physically meaningful: (i) the quantity Δ which is a measure of the saturation in the system:(5)$$\Delta =\frac{C-c_{0}}{c_{0}}=\frac{C}{c_{0}}-1,\;$$
and (ii) rescaled kinetic coefficient $\beta_{0}$:(6)$$\beta_{0}=\beta v_{0}.\;$$
Indeed, rescaling the bare variables R and t to dimensionless quantities r=r(τ) and τ via the relations:(7)$$r=\frac{R}{2\Gamma },\;$$
and
(8)$$\tau =\frac{v_{0}}{2\Gamma }t\;\;$$
is useful for carrying out a solid semi-quantitative description of the spherulite’s formation equation. 

From the system (4)–(8), one derives the following nonlinear differential equation
(9)$$\beta_{0}{\left(\frac{dr}{d\tau }\right)}^{2}+\left(\Delta +\beta_{0}-\frac{1}{r}\right)\frac{dr}{d\tau }-\frac{1}{r}-1=0.$$

Equation (9) is an algebraic quadratic equation with respect to dr/dτ, where x=dr/dτ, which can be rewritten as:(10)$$\beta_{0}x^{2}+\frac{\left(\Delta +\beta_{0}\right)r-1}{r}x-\frac{r+1}{r}=0.$$

The real valued roots of this equation can be determined by using the conventional method of solving quadratic equations, namely, with specifying the characteristic $\Delta_{x}$:(11)$$\Delta_{x}=\frac{{\left[\left(\Delta +\beta_{0}\right)r-1\right]}^{2}+4\beta_{0}r\left(r+1\right)}{r^{2}}>0,$$
and its necessary square root: (12)$$\sqrt{\Delta_{x}}=\frac{\sqrt{{\left[\left(\Delta +\beta_{0}\right)r-1\right]}^{2}+4\beta_{0}r\left(r+1\right)}}{r}:=d\left(r\right)/r,\;$$
thus, if the numerator of the fraction in Equation (12) is denoted for convenience by d(r). The roots of the quadratic equation are explicitly given by:(13)$$x_{1|2}=\frac{-\left[\left(\Delta +\beta_{0}\right)r-1\right]\pm \sqrt{{\left[\left(\Delta +\beta_{0}\right)r-1\right]}^{2}+4\beta_{0}r\left(r+1\right)}}{2\beta_{0}r}.\;$$

One of its roots has to be ruled out. It is determined by the limiting case $\beta_{0}\to 0$. (This, contrary to the vanishing kinetic limit of the phenomenon, can rather be ascribed to its thermodynamic counterpart.) Finally, one gets: (14)$$\frac{dr}{d\tau }=x_{2}=\frac{\sqrt{{\left[\left(\Delta +\beta_{0}\right)r-1\right]}^{2}+4\beta_{0}r\left(r+1\right)}}{2\beta_{0}r}+\frac{1}{2\beta_{0}r}-\frac{\Delta +\beta_{0}}{2\beta_{0}},\;$$
or by employing the shorter notation with d(r)
(15)$$\frac{dr}{d\tau }=\frac{d\left(r\right)+1}{2\beta_{0}r}-\frac{\Delta +\beta_{0}}{2\beta_{0}}>0,\;$$
where explicitly
(16)$$d\left(r\right)=\sqrt{{\left[\left(\Delta +\beta_{0}\right)r-1\right]}^{2}+4\beta_{0}r\left(r+1\right)}.\;$$

Notice that, by presenting above the rescaled spherulite’s evolution equation, that means Equation (15) with its supporting Equation (16), we have arrived at the equation on which we wish to perform a simplified structural stability analysis in order to reveal the onset of the spherulitic growth. 

Numerical solutions of Equation (15) by using Euler’s discretization method are presented in Figure 2. The dependence of r(τ) on the rescaled kinetic $\left(\beta_{0}\right)$ and thermodynamic (Δ) dimensionless parameters indicate the edge of diffusion- and mass-convection-driven growth at the onset of spherulitic formation around $\beta_{0}\approx \Delta \approx 0.2$. A square root tendency manifests earlier for the set of upper curves, thus for smaller values of $\beta_{0}$.

## 3. Results and Discussion

First of all, let us note that, at the onset of the spherulitic growth at which the overall system is settled in front of a decision whether it will evolve either in diffusional or in non-diffusional (mass-convection-like) limit, one can derive the double root ($\Delta_{x}=0$) of the quadratic Equation (10)
(17)$$x_{0}=\frac{1-\left(\Delta +\beta_{0}\right)r}{2\beta_{0}r}=\frac{dr}{d\tau }$$
to be equivalent to
(18)$$\frac{dr}{d\tau }=\frac{1}{2\beta_{0}r}-\frac{\Delta +\beta }{2\beta_{0}}$$
which, after denoting the radius by $r_{d}$, can be named the diffusion limit if
(19)$$\frac{1}{r_{d}}-\Delta +\beta_{0}>0$$
or
(20)$$\frac{1}{r_{d}}>\Delta -\beta_{0}.$$

It implies that the critical value of the non-spherulitic or diffusional growth reads
(21)$$r_{d}=\frac{1}{\Delta -\beta_{0}}.$$

The non-dimensional radius given by Equation (21) reflects an interplay between thermodynamic (Δ) and kinetic ($\beta_{0}$) parts at the onset of the diffusion-driven but non-spherulitic growth. Cleary, one can also expect such mode of growth if one puts d(r)=0 in Equation (15), making then use of its equivalence with Equation (18) with zeroth condition applied to Equation (16). 

The non-diffusive, thus, the spherulitic onset of the evolution, arises if one assumes a proportionality of d(r) to r; thus, when providing a linearity thereof (but not a constancy with non-negative property), namely
(22)$$d\left(r\right)∼r\Rightarrow \;{\left[\left(\Delta +\beta_{0}\right)r-1\right]}^{2}+4\beta_{0}r\left(r+1\right)∼r^{2}$$
which implies that
(23)$${\left(\Delta +\beta_{0}\right)}^{2}r^{2}-2\left(\Delta +\beta_{0}\right)r+1+4\beta_{0}r^{2}+4\beta_{0}r∼r^{2}$$
and a simple limit of the form after postponing all in-r quadratic (counter-balancing) terms
(24)$$2r\left(2\beta_{0}-\Delta -\beta_{0}\right)+1\to 0$$
can ultimately be taken, which results in
(25)$$2r\left(\beta_{0}-\Delta \right)\to -1$$
and eventually leads to (denote the limit by $r_{nd}$)
(26)$$r_{nd}=\frac{1}{2\left(\Delta -\beta_{0}\right)}.$$

It is if $\beta_{0}\approx 0.2$, cf. Figure 2, which is when the kinetics and thermodynamics work at the singularity-expressing onset of the spherulitic formation, with Δ≈0.2, c.f. Equations (21) and (26).

Comparing Equation (26) with Equation (21), one immediately recognizes that a simple relation
(27)$$r_{d}=2\cdot r_{nd}$$
prevails. Despite the physical fact that the factor two in proportionality relation (27) means an earlier onset (as compared to that non-spherulitic viz diffusional mode) of the spherulitic growth to manifest. The factor two can also be interpreted in graphical terms of Figure 1, namely, that in the diffusion (Mullins–Sekerka type) limit characteristic of $r\;\sim t^{1/2}$, the only one crystalline phase builds the object (Figure 1a), while in the case of the spherulitic growth, two concurrent phases (crystalline vs. amorphous) constitute the spherulite’s body, cf. Figure 1b. 

Thus, this way it has been shown that, at the onset of the growth under study and presumably around the kinetic-thermodynamic singularity $\beta_{0}\sim \Delta$, cf. Equations (21) and (26), the spherulitic growth will prevail earlier in terms of the spherulite’s nucleus value, cf. Equation (27), if the mass-convection and nonequilibrium (Goldenfeld type [3]) boundary conditions win over those of the Mullins–Sekerka kind. This feasible singularity limit at the onset of the growth, namely, $\beta_{0}\sim \Delta$, suggests that, though the model is deterministic, its extension can fairly be envisioned towards applying prospectively a stochastic approach [15], wherein the corresponding fluctuations around the $\beta_{0}\sim \Delta$ condition can show up in subtle or pronounced ways. 

The overall spherulitic formation as viewed in the so-called entropy-production (e) terms [16] detected at the interface can be associated with the stochastic-fluctuational context. The scalar product $\vec{j}\left[c\left(R\right)\right]\circ {\vec{n}}_{0}$ put at the r.h.s. of Equation (1) involves the matter flux $\vec{j}\left[c\left(R\right)\right]$ of the external feeding field. The field is twofold, namely, either of mass-convective or of locally diffusional character. In the former, it does not include curvatures, whereas in the latter it receives them for granted, see Figure 1a, and it basically goes like $\vec{j}\left[c\left(R\right)\right]\circ \;{\vec{n}}_{0}\;\sim \;1/r$. For the mass-convective counterpart, one gets $\vec{j}\left[c\left(R\right)\right]\circ {\vec{n}}_{0}\;\sim const$, when late times conditions readily apply. (In fact, in this time zone, the local curvatures of the Mullins–Sekerka mode also cease to grow, yielding ultimately a similar physical scenario.) The entropy production $e=\vec{j}\left[c\left(R\right)\right]\circ \vec{x}$ with $\vec{x}$ representing the (Onsager type) thermodynamic force enables, while based on the same reasoning, to ascertain that the involvement of (local) curvature term and the application of the Fick’s law $\vec{x}\circ {\vec{n}}_{0}\;\sim 1/r$ to e gives a bigger non-negative account to it based thoroughly on the Mullins–Sekerka [13,17] (crystal growth) mode than in the case of mass-convective, thus spherulitic mode, cf. Equation (3). 

## 4. Conclusions

In this study, we have demonstrated, while based on the non-dimensional model (suitable for numerics), that upon mass-convection and nonequilibrium boundary criteria for the (poly)crystal’s growth, such as the one of (bio)polymers addressing or that concerning biominerals (geophysical objects), realized in defects containing and condensed matter involving matrix, that a well-justifiable chance of spherulites’ emergence prior to a pure diffusion-controlled crystal growth exists at the onset of the growing conditions. The argumentation line is based on the physical fact that, in spherulites (polycrystals) two phases may “synergistically” coexist, whereas in single crystals the only ordered crystalline phase has to be built in suitably, presumably at a higher energetic cost than in the former. As named by us, the unimodal crystalline Mullins–Sekerka type mode of growth, characteristic of local curvatures’ presence, seems to be more entropy-productive in its emerging (structural) nature than the so-named bimodal or Goldenfeld type mode of growth in which the local curvatures do not play any crucial roles, and in which kinetics seems to win over thermodynamics. In turn, a liaison of amorphous and crystalline phases makes the system far better compromised to the thermodynamic-kinetic conditions it actually, and concurrently, follows.

The final conclusions presented are based on the peculiar evolution equation, having corroborated cooperatively mass-convective and nonequilibrium (boundary) conditions, that basically drive the growing system far from equilibrium. Interestingly, one may qualitatively predict that the entropy production [16] in such massive (poly)crystalline forms is more expressed in the unimodal non-compromised Mullins–Sekerka type mode. In this mode, a thermodynamic, close-to-equilibrium supersaturation factor prevails, unlike in its Goldenfeld-like bimodal nonequilibrium counterpart. This bimodality, i.e., a synergistic coexistence of crystalline and amorphous phases within a growing spherulite, rests on terms of internal stress–strain material conditions, and does not admit the local curvatures at the interface to prevail. The latter is not the case of any diffusion-controlled (unimodal) growth in which the so-called Mullins–Sekerka instability manifests readily. Finally, let us stress that it seems to us that the non-dimensionality of the proposed modeling suggests that the system does not depend upon experimental details, manifesting somehow a quasi-universal, that means, mainly the scaling addressing character of the performed modeling [18].

## Figures and Tables

**Figure 1 entropy-24-00663-f001:**
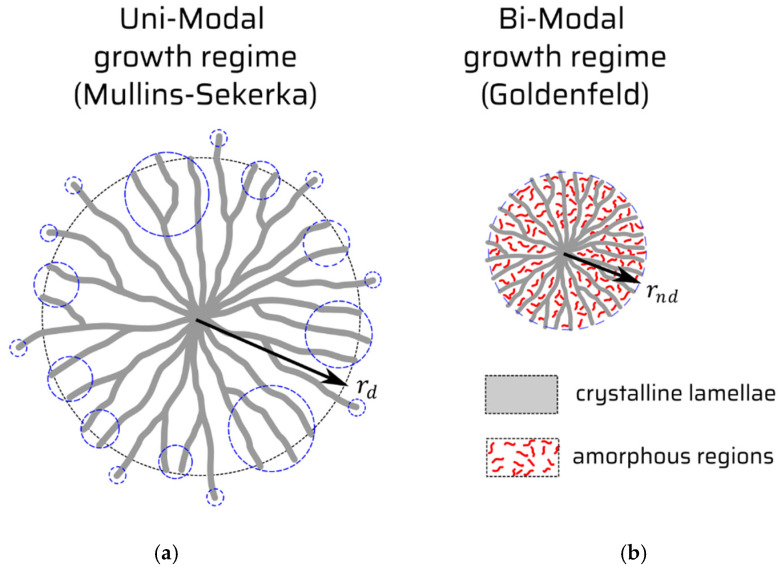
Mullins–Sekerka (M–S) type and Goldenfeld (G) type: (**a**) M–S, local curvatures are indicated by blue dashed circles; (**b**) G, mean curvature is signified by blue dashed circle.

**Figure 2 entropy-24-00663-f002:**
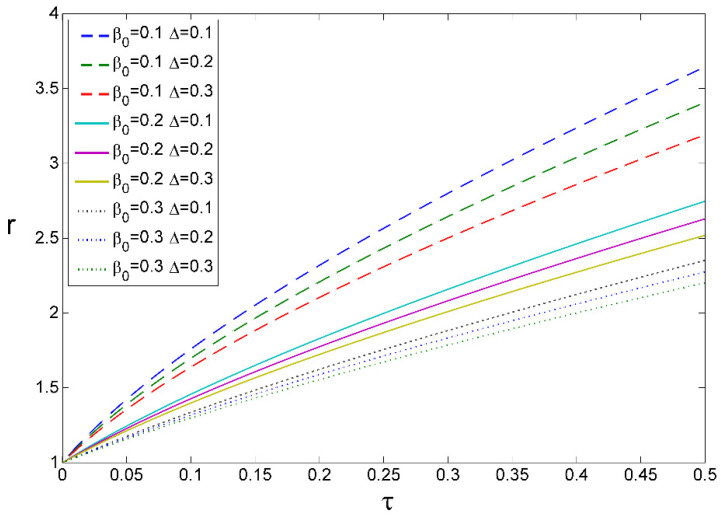
The dependence of r(τ) on the rescaled kinetic $\beta_{0}$ and thermodynamic Δ dimensionless parameters. In the chosen time interval, the curves reflect a visible tendency to pass from the diffusion-like $\left(\beta_{0}\to 0\right)$ to mass-convective-type mode $\left(\beta_{0}↛0\right)$. Other realizations of r(τ) are presented in [12].

## Data Availability

Not applicable.

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
