# Peer review of "Spherulites: How Do They Emerge at an Onset of Nonequilibrium Kinetic-Thermodynamic and Structural Singularity Addressing Conditions?"

_entropy, 2022, doi:10.3390/e24050663_

Round 1

Reviewer 1 Report

Authors study the problem of emergence of spherulite defects in condensed phase with the special attention to change in evolution regime from diffusion-like to wave-like.   The presented manuscript requires some extensions. Thus, I'd reccomend to publish the manuscript after incorporation of some extensions.  

1) The abstract should be rewritten to give a more comprehensive view of obtained results.  

2) Authors try to answer the question why spherulitic evolution changes from diffusion-like to wave-like. This question is related to the dynamical properties of defects growth. Therefore, I'd be expecting some more information on how dynamic properties of defects growth can be measured experimentally.  

3) The difference between diffusion-like and wave-like regimes should be explained more explicitly. Moreover, it should be stated more clearly why obtained solutions correspond to these regimes.  

4)  R should be defined more precisely.  

5) What are the units in Eq. (1)? In general, the problem of units should be discussed in more detail.  

6) A figure presenting the dependence of R(t) should be provided.  

7) A broader discussion of existing methods and results is necessary. For example see: https://doi.org/10.1007/s40828-015-0013-1 http://dx.doi.org/10.1016/j.progpolymsci.2015.11.006

Author Response

Thank you for the evaluation of our manuscript and the valuable comments that were submitted to it.

Based on the comments of the reviewer, the following corrections have been made:

1) The abstract should be rewritten to give a more comprehensive view of obtained results.  

Ad. The abstract has been redrafted as suggested

2) Authors try to answer the question why spherulitic evolution changes from diffusion-like to wave-like. This question is related to the dynamical properties of defects growth. Therefore, I'd be expecting some more information on how dynamic properties of defects growth can be measured experimentally.  

Ad. A relevant fragment has been added in the introduction: “The method of revealing this phenomenon turns out to be the DSC (Differential Scanning Calorimetry) and the underlying process is coalescence of the spherulitic material, also resulting in a structural impingement of the spherulites [6]” where [6] is a new citation:  Raimo, M. Growth of Spherulites: Foundation of the DSC Analysis of Solidification. ChemTexts 2015, 1, 13, doi:10.1007/s40828-015-0013-1.

3) The difference between diffusion-like and wave-like regimes should be explained more explicitly. Moreover, it should be stated more clearly why obtained solutions correspond to these regimes.  

Ad. Appropriate clarifications have been included throughout the text. Some of the inaccuracies related to the term "wave-like" have been changed by changing the nomenclature to "mass convection".

4)  R should be defined more precisely.  

Ad. Appropriate clarifications have been included in the text. R is the spherulite’s radius, see text.

5) What are the units in Eq. (1)? In general, the problem of units should be discussed in more detail.  

Ad. Units have been specified. Both sides of eq. (1) are given SI units of kg⁄(m^2 s).

6) A figure presenting the dependence of R(t) should be provided.  

Ad. The corresponding chart has been added as Figure 2: The dependence of r(τ) on the rescaled kinetic β_0 and thermodynamic ∆ dimensionless parameters.

7) A broader discussion of existing methods and results is necessary. For example see: doi.org/10.1007/s40828-015-0013-1, dx.doi.org/10.1016/j.progpolymsci.2015.11.006

Ad. The papers indicated by the Reviewer were cited in the appropriate places with a commentary.

Reviewer 2 Report

Please, find attached my comments and suggestions for Authors.

Author Response

Thank you for the evaluation of our manuscript and the valuable comments that were submitted to it.

Based on the comments of the reviewer, the following corrections have been made:

1) Scientifically speaking, the article presents an interesting take from the authors on the mechanisms of spherulite growth. I strongly agree with the authors that the spherulitic growth suggests underlying out-of-equilibrium dynamics and, as such, should be studied in the framework of far-from-equilibrium thermodynamics. However, I am not sure to understand what motivates the authors to assume that the growth dynamics are governed by convection. A quick explanation in the introduction on how convection occurs in spherulitic growth would be greatly appreciated.

Ad. Some of the inaccuracies related to the term "wave-like" have been changed by changing the nomenclature to "mass convection". Appropriate clarifications have been included throughout the text, among others: “Amongst many answers to this question, there is at least a pronounced streamline of arguments proclaiming that the growing system of interest is evolving in nonequilibrium thermodynamic boundary conditions [3,7]. Being motivated by afore presented and not answered in full experimental observations, in what follows, we are attempting to provide a simple theoretical rationalization that it is convincingly seen in terms of our type of modeling.”

2) The manuscript contains all expected components (Introduction, Model, Results and Analysis, Conclusion) and is well-developed. The literature is relevant to the subject. Unfortunately, I cannot say that the article is well-written, as I had trouble understanding a lot of sentences. I strongly recommend the authors to shorten and simplify sentences and avoid convoluted explanations.

Ad. The sentences have been shortened and simplified as much as possible.

3) Specific comments

Ad. Specific comments were taken into account, including:

    • Some of the inaccuracies related to the term "wave-like" have been changed by changing the nomenclature to "mass convection".
    • There was an editing error in Equation (24). The first segment, on the left, lacks the power of "2" in r. Everything should be clear now
    • The term "mean-field" has been removed,: “This feasible singularity limit at the onset of the growth, namely β_0~Δ, suggests that though the model is deterministic, its extension can fairly be envisioned towards applying prospectively a stochastic approach [15], wherein the corresponding fluctuations around the β_0~Δ condition can show up in a subtle or pronounced ways.”
    • Dimensionlessness has been clarified in Abstract: “The dimensionless character of the modeling suggests that the system does not directly depend upon experimental details, manifesting somehow its quasi-universal, i.e. scaling addressing character.” And at the end of the conclusion: “And finally, let us stress that it seems to us that the non-dimensionality of the proposed modeling suggests that the system does not depend upon experimental details, manifesting somehow a quasi-universal, that means, mainly scaling addressing character of the performed modeling [18].”